# A KL-LUCB Bandit Algorithm for Large-Scale Crowdsourcing

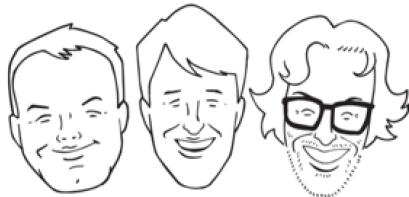

**Ervin Tánczos*** and **Robert Nowak**[†]
University of Wisconsin-Madison
tanczos@wisc.edu,  rdnowak@wisc.edu

**Bob Mankoff**
Former Cartoon Editor of the New Yorker
bmankoff@hearst.com

## Abstract

This paper focuses on best-arm identification in multi-armed bandits with bounded rewards. We develop an algorithm that is a fusion of lil-UCB and KL-LUCB, offering the best qualities of the two algorithms in one method. This is achieved by proving a novel anytime confidence bound for the mean of bounded distributions, which is the analogue of the LIL-type bounds recently developed for sub-Gaussian distributions. We corroborate our theoretical results with numerical experiments based on the New Yorker Cartoon Caption Contest.

## 1 Multi-Armed Bandits for Large-Scale Crowdsourcing

This paper develops a new multi-armed bandit (MAB) for large-scale crowdsourcing, in the style of the KL-UCB [4, 9, 3]. Our work is strongly motivated by crowdsourcing contests, like the New Yorker Cartoon Caption contest [10][3]. The new approach targets the "best-arm identification problem" [1] in the fixed confidence setting and addresses two key limitations of existing theory and algorithms:

(i) State of the art algorithms for best arm identification are based on sub-Gaussian confidence bounds [5] and fail to exploit the fact that rewards are usually bounded in crowdsourcing applications.

(ii) Existing KL-UCB algorithms for best-arm identification do exploit bounded rewards [8] , but have suboptimal performance guarantees in the fixed confidence setting, both in terms of dependence on problem-dependent hardness parameters (Chernoff information) and on the number of arms, which can be large in crowdsourcing applications.

The new algorithm we propose and analyze is called **lil-KLUCB**, since it is inspired by the lil-UCB algorithm [5] and the KL-LUCB algorithm [8]. The lil-UCB algorithm is based on sub-Gaussian bounds and has a sample complexity for best-arm identification that scales as

$$\sum_{i \geq 2} \Delta_i^{-2} \log(\delta^{-1} \log \Delta_i^{-2}) \,,$$

where $\delta \in (0, 1)$ is the desired confidence and $\Delta_i = \mu_1 - \mu_i$ is the gap between the means of the best arm (denoted as arm 1) and arm $i$. If the rewards are in $[0, 1]$, then the KL-LUCB algorithm has

---

[†]This work was partially supported by the NSF grant IIS-1447449 and the AFSOR grant FA9550-13-1-0138.

[3]For more details on the New Yorker Cartoon Caption Contest, see the Supplementary Materials.

a sample complexity scaling essentially like[4]

$$\sum_{i \geq 2}(D_i^*)^{-1} \log(n\delta^{-1}(D_i^*)^{-1}) \,,$$

where $n$ is the number of arms and $D_i^* := D^*(\mu_1, \mu_i)$ is the Chernoff-information between a $\mathrm{Ber}(\mu_1)$ and a $\mathrm{Ber}(\mu_i)$ random variable[5]. Ignoring the logarithmic factor, this bound is optimal for the case of Bernoulli rewards [7, 11]. Comparing these two bounds, we observe that KL-LUCB may offer benefits since $D_i^* = D^*(\mu_1, \mu_i) \geq (\mu_1 - \mu_i)^2/2 = \Delta_i^2/2$, but lil-UCB has better logarithmic dependence on the $\Delta_i^2$ and no explicit dependence on the number of arms $n$. Our new algorithm lil-KLUCB offers the best of both worlds, providing a sample complexity that scales essentially like

$$\sum_{i \geq 2}(D_i^*)^{-1} \log(\delta^{-1} \log(D_i^*)^{-1}) \,.$$

The key to this result is a novel anytime confidence bound for sums of bounded random variables, which requires a significant departure from previous analyses of KL-based confidence bounds.

The practical benefit of lil-KLUCB is illustrated in terms of the New Yorker Caption Contest problem [10]. The goal of that crowdsourcing task is to identify the funniest cartoon caption from a batch of $n \approx 5000$ captions submitted to the contest each week. The crowd provides "3-star" ratings for the captions, which can be mapped to $\{0, 1/2, 1\}$, for example. Unfortunately, many of the captions are not funny, getting average ratings close to $0$ (and consequently very small variances). This fact, however, is ideal for KL-based confidence intervals, which are significantly tighter than those based on sub-Gaussianity and the worst-case variance of $1/4$. Compared to existing methods, the lil-KLUCB algorithm better addresses the two key features in this sort of application: (1) a very large number of arms, and (2) bounded reward distributions which, in many cases, have very low variance. In certain instances, this can have a profound effect on sample complexity (e.g., $O(n^2)$ complexity for algorithms using sub-Gaussian bounds vs. $O(n \log n)$ for lil-KLUCB, as shown in Table 1).

The paper is organized as follows. Section 2 defines the best-arm identification problem, gives the lil-KLUCB algorithm and states the main results. We also briefly review related literature, and compare the performance of lil-KLUCB to that of previous algorithms. Section 3 provides the main technical contribution of the paper, a novel anytime confidence bound for sums of bounded random variables. Section 4 analyzes the performance of the lil-KLUCB algorithm. Section 5 provides experimental support for the lil-KLUCB algorithm using data from the New Yorker Caption Contest.

## 2    Problem Statement and Main Results

Consider a MAB problem with $n$ arms. We use the shorthand notation $[n] := \{1, \ldots, n\}$. For every $i \in [n]$ let $\{X_{i,j}\}_{j \in \mathbb{N}}$ denote the reward sequence of arm $i$, and suppose that $\mathbb{P}(X_{i,j} \in [0,1]) = 1$ for all $i \in [n]$, $j \in \mathbb{N}$. Furthermore, assume that all rewards are independent, and that $X_{i,j} \sim \mathbb{P}_i$ for all $j \in \mathbb{N}$. Let the mean reward of arm $i$ be denoted by $\mu_i$ and assume w.l.o.g. that $\mu_1 > \mu_2 \geq \cdots \geq \mu_n$.

We focus on the best-arm identification problem in the fixed-confidence setting. At every time $t \in \mathbb{N}$ we are allowed to select an arm to sample (based on past rewards) and observe the next element in its reward sequence. Based on the observed rewards, we wish to find the arm with the highest mean reward. In the fixed confidence setting, we prescribe a probability of error $\delta \in (0,1)$ and our goal is to construct an algorithm that finds the best arm with probability at least $1 - \delta$. Among $1 - \delta$ accurate algorithms, one naturally favors those that require fewer samples. Hence proving upper bounds on the sample complexity of a candidate algorithm is of prime importance.

The lil-KLUCB algorithm that we propose is a fusion of lil-UCB [5] and KL-LUCB [8], and its operation is essentially a special instance of LUCB++ [11]. At each time step $t$, let $T_i(t)$ denote the total number of samples drawn from arm $i$ so far, and let $\widehat{\mu}_{i,T_i(t)}$ denote corresponding empirical mean. The algorithm is based on lower and upper confidence bounds of the following general form:

for each $i \in [n]$ and any $\delta \in (0, 1)$

$$L_i(t, \delta) = \inf \left\{ m < \widehat{\mu}_{i,T_i(t)} : \ D\left(\widehat{\mu}_{i,T_i(t)}, m\right) \leq \frac{c \log \left(\kappa \log_2(2T_i(t))/\delta\right)}{T_i(t)} \right\}$$

$$U_i(t, \delta) = \sup \left\{ m > \widehat{\mu}_{i,T_i(t)} : \ D\left(\widehat{\mu}_{i,T_i(t)}, m\right) \leq \frac{c \log \left(\kappa \log_2(2T_i(t))/\delta\right)}{T_i(t)} \right\}$$

where $c$ and $\kappa$ are small constants (defined in the next section). These bounds are designed so that with probability at least $1 - \delta$, $L_i(T_i(t), \delta) \leq \mu_i \leq U_i(T_i(t), \delta)$ holds for all $t \in \mathbb{N}$. For any $t \in \mathbb{N}$ let $\mathrm{TOP}(t)$ be the index of the arm with the highest empirical mean, breaking ties at random. With this notation, we state the lil-KLUCB algorithm and our main theoretical result.

---

**lil-KLUCB**

1. **Initialize** by sampling every arm once.
2. **While** $L_{\mathrm{TOP}(t)}(T_{\mathrm{TOP}(t)}(t), \delta/(n-1)) \leq \max\limits_{i \neq \mathrm{TOP}(t)} U_i(T_i(t), \delta)$ **do:**
   - Sample the following two arms:
     - $\mathrm{TOP}(t)$, and
     - $\arg \max\limits_{i \neq \mathrm{TOP}(t)} U_i(T_i(t), \delta)$

   and update means and confidence bounds.
3. **Output** $\mathrm{TOP}(t)$

---

**Theorem 1.** *For every $i \geq 2$ let $\widetilde{\mu}_i \in (\mu_i, \mu_1)$, and $\widetilde{\mu} = \max_{i \geq 2} \widetilde{\mu}_i$. With probability at least $1 - 2\delta$, lil-KLUCB returns the arm with the largest mean and the total number of samples it collects is upper bounded by*

$$\inf_{\widetilde{\mu}_2, \ldots, \widetilde{\mu}_n} \frac{c_0 \log \left((n-1)\delta^{-1} \log D^*(\mu_1, \widetilde{\mu})^{-1}\right)}{D^*(\mu_1, \widetilde{\mu})} + \sum_{i \geq 2} \frac{c_0 \log \left(\delta^{-1} \log D^*(\mu_i, \widetilde{\mu}_i)^{-1}\right)}{D^*(\mu_i, \widetilde{\mu}_i)} ,$$

*where $c_0$ is some universal constant, $D^*(x, y)$ is the Chernoff-information.*

**Remark 1.** *Note that the LUCB++ algorithm of [11] is general enough to handle identification of the top $k$ arms (not just the best-arm). All arguments presented in this paper also go through when considering the top-$k$ problem for $k > 1$. However, to keep the arguments clear and concise, we chose to focus on the best-arm problem only.*

## 2.1 Comparison with previous work

We now compare the sample complexity of lil-KLUCB to that of the two most closely related algorithms, KL-LUCB [8] and lil-UCB [5]. For a detailed review of the history of MAB problems and the use of KL-confidence intervals for bounded rewards, we refer the reader to [3, 9, 4].

For the KL-LUCB algorithm, Theorem 3 of [8] guarantees a high-probability sample complexity upper bound scaling as

$$\inf_{c \in (\mu_1, \mu_2)} \sum_{i \geq 1} (D^*(\mu_i, c))^{-1} \log \left(n\delta^{-1}(D^*(\mu_i, c))^{-1}\right) .$$

Our result improves this in two ways. On one hand, we eliminate the unnecessary logarithmic dependence on the number of arms $n$ in every term. Note that the $\log n$ factor still appears in Theorem 1 in the term corresponding to the number of samples on the best arm. It is shown in [11] that this factor is indeed unavoidable. The other improvement lil-KLUCB offers over KL-LUCB is improved logarithmic dependence on the Chernoff-information terms. This is due to the tighter confidence intervals derived in Section 3.

Comparing Theorem 1 to the sample complexity of lil-UCB, we see that the two are of the same form, the exception being that the Chernoff-information terms take the place of the squared mean-gaps

(which arise due to the use of sub-Gaussian (SG) bounds). To give a sense of the improvement this can provide, we compare the sums[6]

$$S_{\mathrm{KL}} = \sum_{i \geq 2} \frac{1}{D^*(\mu_i, \mu_1)} \quad \text{and} \quad S_{\mathrm{SG}} = \sum_{i \geq 2} \frac{1}{\Delta_i^2} \ .$$

Let $\mu, \mu' \in (0, 1)$, $\mu < \mu'$ and $\Delta = |\mu - \mu'|$. Note that the Chernoff-information between $\mathrm{Ber}(\mu)$ and $\mathrm{Ber}(\mu')$ can be expressed as

$$D^*(\mu, \mu') = \max_{x \in [\mu, \mu']} \min\{D(x, \mu), D(x, \mu')\} = D(x^*, \mu) = D(x^*, \mu') = \frac{D(x^*, \mu) + D(x^*, \mu')}{2} \ ,$$

for some unique $x^* \in [\mu, \mu']$. It follows that

$$D^*(\mu, \mu') \geq \min_{x \in [\mu, \mu']} \frac{D(x, \mu) + D(x, \mu')}{2} = \log \frac{1}{\sqrt{\mu(\mu + \Delta)} + \sqrt{(1 - \mu)(1 - \mu - \Delta)}} \ .$$

Using this with every term in $S_{\mathrm{KL}}$ gives us an upper bound on that sum. If the means are all bounded well away from 0 and 1, then $S_{\mathrm{KL}}$ may not differ that much from $S_{\mathrm{SG}}$. There are some situations however, when the two expressions behave radically differently. As an example, consider a situation when $\mu_1 = 1$. In this case we get

$$S_{\mathrm{KL}} \leq \sum_{i \geq 2} \frac{2}{\log \frac{1}{1 - \Delta_i}} \leq 2 \sum_{i \geq 2} \frac{1}{\Delta_i} \ll \sum_{i \geq 2} \frac{1}{\Delta_i^2} = S_{\mathrm{SG}} \ .$$

Table 1 illustrates the difference between the scaling of the sums $S_{\mathrm{KL}}$ and $S_{\mathrm{SG}}$ when the gaps have the parametric form $\Delta_i = (i/n)^\alpha$.

Table 1: $S_{\mathrm{KL}}$ versus $S_{\mathrm{SG}}$ for mean gaps $\Delta_i = (\frac{i}{n})^\alpha$, $i = 1, \ldots, n$

| $\alpha$ | $\in (0, 1/2)$ | $1/2$ | $\in (1/2, 1)$ | $1$ | $\in (1, \infty)$ |
|---|---|---|---|---|---|
| $S_{\mathrm{KL}}$ | $n$ | $n$ | $n$ | $n \log n$ | $n^\alpha$ |
| $S_{\mathrm{SG}}$ | $n$ | $n \log n$ | $n^{2\alpha}$ | $n^2$ | $n^{2\alpha}$ |

We see that KL-type confidence bounds can sometimes provide a significant advantage in terms of the sample complexity. Intuitively, the gains will be greatest when many of the means are close to 0 or 1 (and hence have low variance). We will illustrate in Section 5 that such gains often also manifest in practical applications like the New Yorker Caption Contest problem.

## 3 Anytime Confidence Intervals for Sums of Bounded Random Variables

The main step in our analysis is proving a sharp anytime confidence bound for the mean of bounded random variables. These will be used to show, in Section 4, that lil-KLUCB draws at most $O((D_i^*)^{-1} \log \log(D_i^*)^{-1})$ samples from a suboptimal arm $i$, where $D_i^* := D^*(\mu_1, \mu_i)$ is the Chernoff-information between a $\mathrm{Ber}(\mu_1)$ and a $\mathrm{Ber}(\mu_i)$ random variable and arm 1 is the arm with the largest mean. The iterated log factor is a necessary consequence of the law-of-the-iterated logarithm [5], and in it is in this sense that we call the bound sharp. Prior work on MAB algorithms based on KL-type confidence bounds [4, 9, 3] did not focus on deriving tight anytime confidence bounds.

Consider a sequence of iid random variables $Y_1, Y_2, \ldots$ that are bounded in $[0, 1]$ and have mean $\mu$. Let $\widehat{\mu}_t = \frac{1}{t} \sum_{j \in [t]} Y_j$ be the empirical mean of the observations up to time $t \in \mathbb{N}$.

**Theorem 2.** *Let $\mu \in [0, 1]$ and $\delta \in (0, 1)$ be arbitrary. Fix any $l \geq 0$ and set $N = 2^l$, and define*

$$\kappa(N) = \delta^{1/(N+1)} \left( \sum_{t \in [N]} \mathbf{1}_{\{l \neq 0\}} \log_2(2t)^{-\frac{N+1}{N}} + N \sum_{k \geq l} (k + 1)^{-\frac{N+1}{N}} \right)^{\frac{N}{N+1}} \ .$$

*(i) Define the sequence $z_t \in (0, 1 - \mu]$, $t \in \mathbb{N}$ such that*

$$D\left(\mu + \tfrac{N}{N+1}z_t, \mu\right) = \frac{\log\left(\kappa(N)\log_2(2t)/\delta\right)}{t} \, , \tag{1}$$

*if a solution exists, and $z_t = 1 - \mu$ otherwise. Then $\mathbb{P}\left(\exists t \in \mathbb{N}: \ \widehat{\mu}_t - \mu > z_t\right) \leq \delta$.*

*(ii) Define the sequence $z_t > 0$, $t \in \mathbb{N}$ such that*

$$D\left(\mu - \tfrac{N}{N+1}z_t, \mu\right) = \frac{\log\left(\kappa(N)\log_2(2t)/\delta\right)}{t} \, ,$$

*if a solution exists, and $z_t = \mu$ otherwise. Then $\mathbb{P}\left(\exists t \in \mathbb{N}: \ \widehat{\mu}_t - \mu < -z_t\right) \leq \delta$.*

The result above can be used to construct anytime confidence bounds for the mean as follows. Consider part *(i)* of Theorem 2 and fix $\mu$. The result gives a sequence $z_t$ that upper bounds the deviations of the empirical mean. It is defined through an equation of the form $D(\mu + Nz_t/(N+1), \mu) = f_t$. Note that the arguments of the function on the left must be in the interval $[0, 1]$, in particular $Nz_t/(N+1) < 1 - \mu$, and the maximum of $D(\mu + x, \mu)$ for $x > 0$ is $D(1, \mu) = \log \mu^{-1}$. Hence, equation 1 does not have a solution if $f_t$ is too large (that is, if $t$ is small). In these cases we set $z_t = 1 - \mu$. However, since $f_t$ is decreasing, equation 1 does have a solution when $t \geq T$ (for some $T$ depending on $\mu$), and this solution is unique (since $D(\mu + x, \mu)$ is strictly increasing).

With high probability $\widehat{\mu}_t - \mu \leq z_t$ for all $t \in \mathbb{N}$ by Theorem 2. Furthermore, the function $D(\mu + x, \mu)$ is increasing in $x \geq 0$. By combining these facts we get that with probability at least $1 - \delta$

$$D\left(\mu + \tfrac{N}{N+1}z_t, \mu\right) \ \geq \ D\left(\tfrac{N\widehat{\mu}_t + \mu}{N+1}, \mu\right) \, .$$

On the other hand

$$D\left(\mu + \tfrac{N}{N+1}z_t, \mu\right) \ \leq \ \frac{\log\left(\kappa(N)\log_2(2t)/\delta\right)}{t} \, ,$$

by definition. Chaining these two inequalities leads to the lower confidence bound

$$L(t, \delta) = \inf\left\{m < \widehat{\mu}_t: \ D\left(\tfrac{N\widehat{\mu}_t + m}{N+1}, m\right) \leq \frac{\log\left(\kappa(N)\log_2(2t)/\delta\right)}{t}\right\} \tag{2}$$

which holds for all times $t$ with probability at least $1 - \delta$. Considering the left deviations of $\widehat{\mu}_t - \mu$ we can get an upper confidence bound in a similar manner:

$$U(t, \delta) = \sup\left\{m > \widehat{\mu}_t: \ D\left(\tfrac{N\widehat{\mu}_t + m}{N+1}, m\right) \leq \frac{\log\left(\kappa(N)\log_2(2t)/\delta\right)}{t}\right\} \, . \tag{3}$$

That is, for all times $t$, with probability at least $1 - 2\delta$ we have $L(t, \delta) \ \leq \ \widehat{\mu}_t \ \leq \ U(t, \delta)$.

Note that the constant $\log \kappa(N) \approx 2\log_2(N)$, so the choice of $N$ plays a relatively mild role in the bounds. However, we note here that if $N$ is sufficiently large, then $\frac{N\widehat{\mu}_t + m}{N+1} \approx \widehat{\mu}_t$, and thus $D\left(\frac{N\widehat{\mu}_t + m}{N+1}, m\right) \approx D(\widehat{\mu}_t, m)$, in which case the bounds above are easily compared to those in prior works [4, 9, 3]. We make this connection more precise and show that the confidence intervals defined as

$$L'(t, \delta) = \inf\left\{m < \widehat{\mu}_t: \ D(\widehat{\mu}_t, m) \leq \frac{c(N)\log\left(\kappa(N)\log_2(2t)/\delta\right)}{t}\right\} \, , \text{ and}$$

$$U'(t, \delta) = \inf\left\{m > \widehat{\mu}_t: \ D(\widehat{\mu}_t, m) \leq \frac{c(N)\log\left(\kappa(N)\log_2(2t)/\delta\right)}{t}\right\} \, ,$$

satisfy $L'(t, \delta) \ \leq \ \widehat{\mu}_t \ \leq \ U'(t, \delta)$ for all $t$, with probability $1 - 2\delta$. The constant $c(N)$ is defined in Theorem 1 in the Supplementary Material, where the correctness of $L'(t, \delta)$ and $U'(t, \delta)$ is shown.

*Proof of Theorem 2.* The proofs of parts *(i)* and *(ii)* are completely analogous, hence in what follows we only prove part *(i)*. Note that $\{\widehat{\mu}_t - \mu > z_t\} \iff \{S_t > tz_t\}$, where $S_t = \sum_{j \in [t]}(Y_j - \mu)$ denotes the centered sum up to time $t$. We start with a simple union bound

$$\mathbb{P}\left(\exists t \in \mathbb{N}: \ S_t > tz_t\right) \leq \mathbb{P}\left(\exists t \in [N]: \ S_t > tz_t\right) + \sum_{k \geq l}\mathbb{P}\left(\exists t \in [2^k, 2^{k+1}]: \ S_t > tz_t\right) \, . \tag{4}$$

First, we bound each summand in the second term individually. In an effort to save space, we define the event $A_k = \{\exists t \in [2^k, 2^{k+1}] : S_t > t z_t\}$. Let $t_{j,k} = (1 + \frac{j}{N}) 2^k$. In what follows we use the notation $t_j \equiv t_{j,k}$. We have

$$\mathbb{P}(A_k) \leq \sum_{j \in [N]} \mathbb{P}\left(\exists t \in [t_{j-1}, t_j] : \ S_t > t z_t\right) \leq \sum_{j \in [N]} \mathbb{P}\left(\exists t \in [t_{j-1}, t_j] : \ S_t > t_{j-1} z_{t_{j-1}}\right) ,$$

where the last step is true if $t z_t$ is non-decreasing in $t$. This technical claim is formally shown in Lemma 1 in the Supplementary Material. However, to give a short heuristic, it is easy to see that $t z_t$ has an increasing lower bound. Noting that $D(\mu + x, \mu)$ is convex in $x$ (the second derivative is positive), and that $D(\mu, \mu) = 0$, we have $D(1, \mu) x \geq D(\mu + x, \mu)$. Hence $z_t \gtrsim t^{-1} \log \log t$.

Using a Chernoff-type bound together with Doob's inequality, we can continue as

$$\mathbb{P}(A_k) \leq \inf_{\lambda > 0} \sum_{j \in [N]} \mathbb{P}\left(\exists t \in [t_{j-1}, t_j] : \ \exp(\lambda S_t) > \exp\left(\lambda t_{j-1} z_{t_{j-1}}\right)\right)$$

$$\leq \sum_{j \in [N]} \exp\left(-\sup_{\lambda > 0}\left(\lambda t_{j-1} z_{t_{j-1}} - \log \mathbb{E}\left(e^{\lambda S_{t_j}}\right)\right)\right)$$

$$= \sum_{j \in [N]} \exp\left(-t_j \sup_{\lambda \geq 0}\left(\lambda \tfrac{N+j-1}{N+j} z_{t_{j-1}} - \log \mathbb{E}\left(e^{\lambda(Y_1 - \mu)}\right)\right)\right) . \tag{5}$$

Using $\mathbb{E}(e^{\lambda Y_1}) \leq \mathbb{E}(e^{\lambda \xi})$ where $\xi \sim \text{Ber}(\mu)$ (see Lemma 9 of [4]), and the notation $\alpha_j = \frac{N+j-1}{N+j}$,

$$\mathbb{P}(A_k) \leq \sum_{j \in [N]} \exp\left(-t_j \sup_{\lambda \geq 0}\left(\lambda \alpha_j z_{t_{j-1}} - \log \mathbb{E}\left(e^{\lambda(\xi - \mu)}\right)\right)\right)$$

$$= \sum_{j \in [N]} \exp\left(-t_j D\left(\mu + \alpha_j z_{t_{j-1}}, \mu\right)\right) , \tag{6}$$

since the rate function of a Bernoulli random variable can be explicitly computed, namely we have $\sup_{\lambda > 0}(\lambda x - \log \mathbb{E}(e^{\lambda \xi})) = D(\mu + x, \mu)$ (see [2]).

Again, we use the convexity of $D(\mu + x, \mu)$. For any $\alpha \in (0, 1)$ we have $\alpha D(\mu + x, \mu) \geq D(\mu + \alpha x, \mu)$, since $D(\mu, \mu) = 0$. Using this with $\alpha = \frac{N}{\alpha_j(N+1)}$ and $x = \alpha_j z_{t_{j-1}}$, we get that

$$\frac{N}{\alpha_j(N+1)} D\left(\mu + \alpha_j z_{t_{j-1}}, \mu\right) \geq D\left(\mu + \tfrac{N}{N+1} z_{t_{j-1}}, \mu\right) .$$

This implies

$$\mathbb{P}(A_k) \leq \sum_{j \in [N]} \exp\left(-t_j \tfrac{N+1}{N} \alpha_j D\left(\mu + \tfrac{N}{N+1} z_{t_{j-1}}, \mu\right)\right) . \tag{7}$$

Plugging in the definition of $t_j$ and the sequence $z_t$, and noting that $\delta < 1$, we arrive at the bound

$$\mathbb{P}(A_k) \leq \sum_{j \in [N]} \exp\left(-\frac{N+1}{N} \log\left(\kappa(N) \log_2(2^{k+1} \tfrac{N+j-1}{N}) / \delta\right)\right) \leq N \left(\frac{\delta}{\kappa(N)(k+1)}\right)^{\frac{N+1}{N}} .$$

Regarding the first term in (4), again using the Bernoulli rate function bound we have

$$\mathbb{P}\left(\exists t \in [N] : \ \widehat{\mu}_t - \mu > z_t\right) \leq \sum_{t \in [N]} \mathbb{P}\left(\widehat{\mu}_t - \mu > z_t\right) \leq \sum_{t \in [N]} \exp\left(-t D(\mu + z_t, \mu)\right) .$$

Using the convexity of $D(\mu + x, \mu)$ as before, we can continue as

$$\mathbb{P}\left(\exists t \in [N] : \ \widehat{\mu}_t - \mu > z_t\right) \leq \sum_{t \in [N]} \exp\left(-t \tfrac{N+1}{N} D\left(\mu + \tfrac{N}{N+1} z_t, \mu\right)\right)$$

$$\leq \sum_{t \in [N]} \exp\left(-\tfrac{N+1}{N} \log\left(\kappa(N) \log_2(2t) / \delta\right)\right)$$

$$\leq \delta^{\frac{N+1}{N}} \kappa(N)^{-\frac{N+1}{N}} \sum_{t \in [N]} \log_2(2t)^{-\frac{N+1}{N}} .$$

Plugging the two bounds back into (4) we conclude that

$$\mathbb{P}\left(\exists t:\ \widehat{\mu}_t - \mu > z_t\right) \le \delta^{\frac{N+1}{N}} \kappa(N)^{-\frac{N+1}{N}} \sum_{j\in[N]} \left(\mathbf{1}_{\{l\neq 0\}} \log_2(2j)^{-\frac{N+1}{N}} + \sum_{k\geq l}(k+1)^{-\frac{N+1}{N}}\right) \le \delta\,,$$

by the definition of $\kappa(N)$. $\hfill\square$

## 4   Analysis of lil-KLUCB

Recall that the lil-KLUCB algorithm uses confidence bounds of the form $U_i(t,\delta) = \sup\{m > \widehat{\mu}_t : D(\widehat{\mu}_t, m) \le f_t(\delta)\}$ with some decreasing sequence $f_t(\delta)$. In this section we make this dependence explicit, and use the notations $U_i(f_t(\delta))$ and $L_i(f_t(\delta))$ for upper and lower confidence bounds. For any $\epsilon > 0$ and $i \in [n]$, define the events $\Omega_i(\epsilon) = \{\forall t \in \mathbb{N}:\ \mu_i \in [L_i(f_t(\epsilon)), U_i(f_t(\epsilon))]\}$.

The correctness of the algorithm follows from the correctness of the individual confidence intervals, as is usually the case with LUCB algorithms. This is shown formally in Proposition 1 provided in the Supplementary Materials. The main focus in this section is to show a high probability upper bound on the sample complexity. This can be done by combining arguments frequently used for analyzing LUCB algorithms and those used in the analysis of the lil-UCB [5]. The proof is very similar in spirit to that of the LUCB++ algorithm [11]. Due to spatial restrictions, we only provide a proof sketch here, while the detailed proof is provided in the Supplementary Materials.

*Proof sketch of Theorem 1.* Observe that at each time step two things can happen (apart from stopping): *(1)* Arm 1 is not sampled (two sub-optimal arms are sampled); *(2)* Arm 1 is sampled together with some other (suboptimal) arm. Our aim is to upper bound the number of times any given arm is sampled for either of the reasons above. We do so by conditioning on the event

$$\Omega' = \Omega_1(\delta) \cap \left(\bigcap_{i\geq 2}\Omega_i(\delta_i)\right)\ ,\ \text{for a certain choice of } \{\delta_i\} \text{ defined below.}$$

For instance, if arm 1 is not sampled at a given time $t$, we know that $\mathrm{TOP}(t) \neq 1$, which means there must be an arm $i \geq 2$ such that $U_i(T_i(t),\delta) \geq U_1(T_1(t),\delta)$. However, on the event $\Omega_1(\delta)$, the UCB of arm 1 is accurate, implying that $U_i(T_i(t),\delta) \geq \mu_1$. This implies that $T_i(t)$ can not be too big, since on $\Omega_i(\delta_i)$, $\widehat{\mu}_{i,t}$ is "close" to $\mu_i$, and also $U_i(T_i(t),\delta)$ is not much larger then $\widehat{\mu}_i$. All this is made formal in Lemma 2, yielding the following upper bound on number of times arm $i$ is sampled for reason *(1)*:

$$\tau_i(\delta \cdot \delta_i) = \min\left\{t \in \mathbb{N}:\ f_t(\delta \cdot \delta_i) < D^*(\mu_i, \mu_1)\right\}\ .$$

Similar arguments can be made about the number of samples of any suboptimal arm $i$ for reason *(2)*, and also the number of samples on arm 1. This results in the sample complexity upper bound

$$\frac{K_1 \log\left((n-1)\delta^{-1}\log D^*(\mu_1,\widetilde{\mu})^{-1}\right)}{D^*(\mu_1,\widetilde{\mu})} + \sum_{i\geq 2}\frac{K_1 \log\left(\delta^{-1}\log D^*(\mu_i,\widetilde{\mu}_i)^{-1}\right) + \log\delta_i^{-1}}{D^*(\mu_i,\widetilde{\mu}_i)}\,,$$

on the event $\Omega'$, where $K_1$ is a universal constant. Finally, we define the quantities $\delta_i = \sup\{\epsilon > 0 : U_i(f_t(\epsilon)) \geq \mu_i\ \forall t \in \mathbb{N}\}$. Note that we have $\mathbb{P}(\delta_i < \gamma) = \mathbb{P}(\exists t \in \mathbb{N}:\ U_i(f_t(\gamma)) \geq \mu_i) \leq \gamma$ according to Theorem 1 in the Supplementary Material. Substituting $\gamma = \exp(-D^*(\mu_i,\widetilde{\mu}_i)z)$ we get

$$\mathbb{P}\left(\frac{\log\delta_i^{-1}}{D^*(\mu_i,\widetilde{\mu}_i)} \geq z\right) \leq \exp(-D^*(\mu_i,\widetilde{\mu}_i)z)\,.$$

Hence $\{\delta_i\}_{i\geq 2}$ are independent sub-exponential variables, which allows us to control their contribution to the sum above using standard techniques. $\hfill\square$

## 5   Real-World Crowdsourcing

We now compare the performance of **lil-KLUCB** to that of other algorithms in the literature. We do this using both synthetic data and real data from the New Yorker Cartoon Caption contest [10][7]. To

keep comparisons fair, we run the same UCB algorithm for all the competing confidence bounds. We set $N = 8$ and $\delta = 0.01$ in our experiments. The confidence bounds are **[KL]**: the KL-bound derived based on Theorem 2, **[SG1]**: a matching sub-Gaussian bound derived using the proof of Theorem 2, using sub-Gaussian tails instead of the KL rate-function (the exact derivations are in the Supplementary Material), and **[SG2]**: the sharper sub-Gaussian bound provided by Theorem 8 of [7].

We compare these methods by computing the empirical probability that the best-arm is among the top 5 empirically best arms, as a function of the total number of samples. We do so using using synthetic data in Figure 5 , where the Bernoulli rewards simulate cases from Table 1, and using real human response data from two representative New Yorker caption contests in Figure 5.

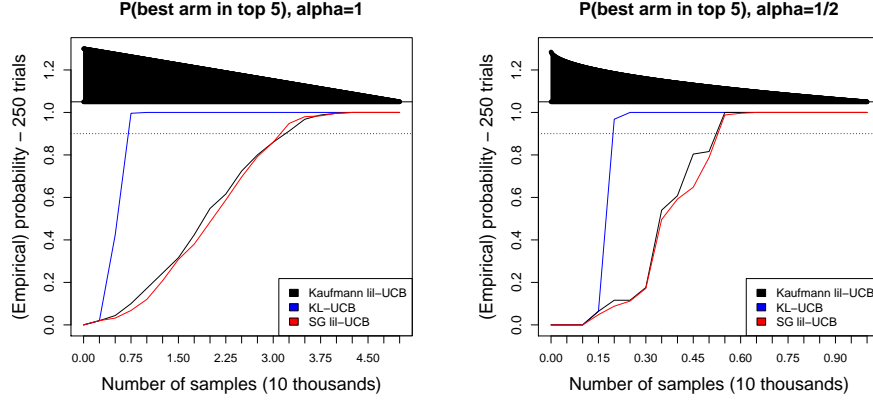

Figure 1: Probability of the best-arm in the top 5 empirically best arms, as a function of the number of samples, based on 250 repetitions. $\mu_i = 1 - ((i-1)/n)^\alpha$, with $\alpha = 1$ in the left panel, and $\alpha = 1/2$ in the right panel. The mean-profile is shown above each plot. **[KL]** Blue; **[SG1]** Red; **[SG2]** Black.

As seen in Table 1, the KL confidence bounds have the potential to greatly outperform the sub-Gaussian ones. To illustrate this indeed translates into superior performance, we simulate two cases, with means $\mu_i = 1 - ((i-1)/n)^\alpha$, with $\alpha = 1/2$ and $\alpha = 1$, and $n = 1000$. As expected, the KL-based method requires significantly fewer samples (about 20 % for $\alpha = 1$ and 30 % for $\alpha = 1/2$) to find the best arm. Furthermore, the arms with means below the median are sampled about 15 and 25 % of the time respectively – key in crowdsourcing applications, since having participants answer fewer irrelevant (and potentially annoying) questions improves both efficiency and user experience.

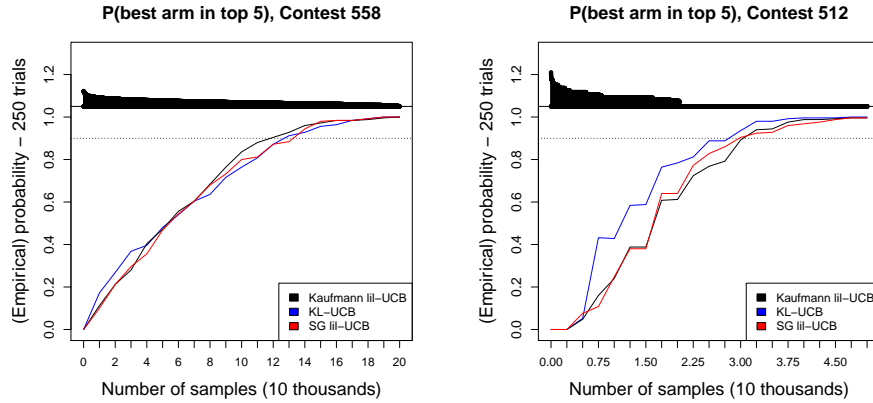

Figure 2: Probability of the best-arm in the top 5 empirically best arms vs. number of samples, based on 250 bootstrapped repetitions. Data from New Yorker contest 558 ($\mu_1 = 0.536$) on left, and contest 512 ($\mu_1 = 0.8$) on right. Mean-profile above each plot. **[KL]** Blue; **[SG1]** Red; **[SG2]** Black.

To see how these methods fair on real data, we also run these algorithms on bootstrapped human response data from the real New Yorker Caption Contest. The mean reward of the best arm in these contests is usually between $0.5$ and $0.85$, hence we choose one contest from each end of this spectrum. At the lower end of the spectrum, the three methods fair comparably. This is expected because the sub-Gaussian bounds are relatively good for means about $0.5$. However, in cases where the top mean is significantly larger than $0.5$ we see a marked improvement in the KL-based algorithm.

## Extension to numerical experiments

Since a large number of algorithms have been proposed in the literature for best arm identification, we include another algorithm in the numerical experiments for comparison.

Previously we compared lil-KLUCB to lil-UCB as a comparison for two reasons. First, this comparison illustrates best the gains of using the novel anytime confidence bounds as opposed to those using sub-Gaussian tails. Second, since lil-UCB is the state of the art algorithm, any other algorithm will likely perform worse.

The authors of [6] compare a number of different best arm identification methods, and conclude that two of them seem to stand out: lil-UCB and Thompson sampling. Therefore, we now include Thomspon sampling **[Th]** in our numerical experiments for the New Yorker data.

We implemented the method as prescribed in [6]. As can bee seen in Figure 5, Thompson sampling seems to perform somewhat worse than the previous methods in these two instances.

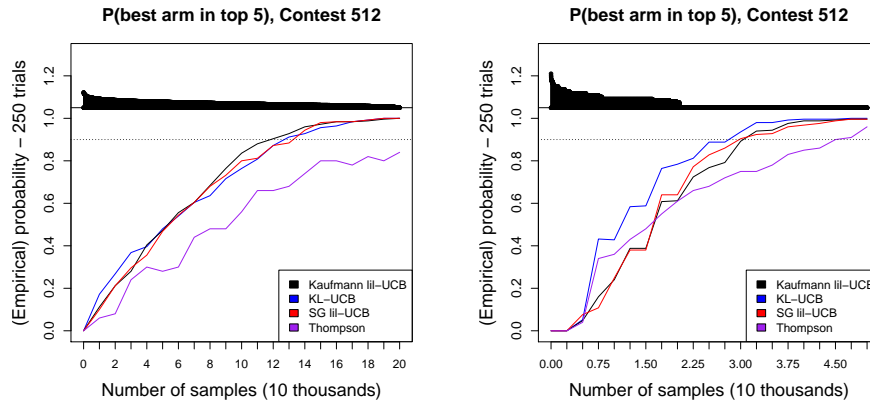

Figure 3: Probability of the best-arm in the top 5 empirically best arms vs. number of samples, based on $250$ bootstrapped repetitions. Data from New Yorker contest 558 ($\mu_1 = 0.536$) on left, and contest 512 ($\mu_1 = 0.8$) on right. Mean-profile above each plot. **[KL]** Blue; **[SG1]** Red; **[SG2]** Black; **[Th]** Purple.

## Footnotes

[4]A more precise characterization of the sample complexity is given in Section 2.

[5]The Chernoff-information between random variables $\mathrm{Ber}(x)$ and $\mathrm{Ber}(y)$ $(0 < x < y < 1)$ is $D^*(x,y) = D(z^*,x) = D(z^*,y)$, where $D(z,x) = z \log \frac{z}{x} + (1-z) \log \frac{1-z}{1-x}$ and $z^*$ is the unique $z \in (x,y)$ such that $D(z,x) = D(z,y)$.

[6]Consulting the proof of Theorem 1 it is clear that the number of samples on the sub-optimal arms of lil-KLUCB scales essentially as $S_{\mathrm{KL}}$ w.h.p. (ignoring doubly logarithmic terms), and a similar argument can be made about lil-UCB. This justifies considering these sums in order to compare lil-KLUCB and lil-UCB.

[7]These data can be found at `https://github.com/nextml/caption-contest-data`

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
