[Supplementary Material · KLUCB_nips_2017_Supplement.pdf]

# Supplementary Materials for "A KL-LUCB Bandit Algorithm for Large-Scale Crowdsourcing"

## Proofs for Section 3

**Lemma 1.** *Let $T$ be the first time index such that (1) has a solution. Since $z_t = 1 - \mu$ by definition for $t < T$, clearly $tz_t$ is increasing for $t \in [T-1]$.*

*Now consider the case $t \geq T - 1$. Using the convexity of $D(\mu + x, \mu)$ (in $x$) and the definition of the sequence $z_t$, we have*

$$D\left(\mu + \tfrac{t}{t+1}\tfrac{N}{N+1}z_t, \mu\right) \leq \tfrac{t}{t+1}D\left(\mu + \tfrac{N}{N+1}z_t, \mu\right)$$
$$\leq \frac{t}{t+1}\frac{\log\left(\kappa(N)\log_2(2t)/\delta\right)}{t}$$
$$\leq \frac{\log\left(\kappa(N)\log_2(2(t+1))/\delta\right)}{t+1}$$
$$= D\left(\mu + \tfrac{N}{N+1}z_{t+1}, \mu\right),$$

*where the last equality holds, since $t \geq T - 1$. Comparing the two ends of this chain of inequalities implies that $\frac{t}{t+1}z_t \leq z_{t+1}$ since the function $D(\mu + x, \mu)$ is increasing in $x$.*

**Theorem 1.** *Consider the setting of Theorem 2 and let*

$$c(N) = \frac{N+1}{N - \log(N+1)} \ .$$

*Define $z_t$ as the solution of*

$$D(\mu + z_t, \mu) = \frac{c(N)\log\left(\kappa(N)\log_2(2t)/\delta\right)}{t} \ ,$$

*if a solution exists, and $z_t = 1 - \mu$ otherwise. Then*

$$(i) : \mathbb{P}\left(\exists t \in \mathbb{N} : \widehat{\mu}_t - \mu > z_t\right) \leq \delta \ ,$$
$$(ii) : \mathbb{P}\left(\exists t \in \mathbb{N} : \widehat{\mu}_t - \mu < -z_t\right) \leq \delta \ .$$

The correctness of the confidence intervals $L'(t, \delta)$ and $U'(t, \delta)$ follow from Theorem 1 in the same way as that of $L(t, \delta)$ and $U(t, \delta)$ follow from Theorem 2 shown in Section 3.

*Proof of Theorem 1.* It is clear by consulting the proof of Theorem 2 that if we had

$$D(\mu + x, \mu) \leq c(N)D\left(\mu + \tfrac{N}{N+1}x, \mu\right) \quad \forall x \in [0, 1-\mu], \ \forall \mu \in (0,1) \ ,$$

then using it at step (7) would yield the desired result.

Let $\alpha \in (0,1)$ and use the notation $D(\mu + x, \mu) = f_\mu(x)$. We wish to show that

$$g_\mu(x) := f_\mu(x) - cf_\mu(\alpha x) \leq 0 \quad \forall x \in [0, 1-\mu], \ \forall \mu \in (0,1) \ .$$

with $c = \frac{1}{\alpha + (1-\alpha)\log(1-\alpha)}$.

We first examine $g_\mu(x)$ as a function of $x$. Recall that the first and second derivatives of $f_\mu(x)$ (in $x$) are

$$f'_\mu(x) = \log\frac{\mu+x}{\mu} - \log\frac{1-\mu-x}{1-\mu} \ , \text{ and } f''_\mu(x) = \frac{1}{(\mu+x)(1-\mu-x)} \ .$$

Hence

$$g''_\mu(x) = f''_\mu(x) - c\alpha^2 f''_\mu(\alpha x) \ .$$

As for the sign of the second derivative, we have

$$f''_\mu(x) - c\alpha^2 f''_\mu(\alpha x) \lesseqgtr 0$$

$$\Updownarrow$$

$$f''_\mu(x) \lesseqgtr c\alpha^2 f''_\mu(\alpha x)$$

$$\Updownarrow$$

$$(\mu + \alpha x)(1 - \mu - \alpha x) \lesseqgtr c\alpha^2(\mu + x)(1 - \mu - x)$$

$$\Updownarrow$$

$$(c-1)\alpha^2 x^2 + \alpha(1-2\mu)(1-c\alpha)x + \mu(1-\mu)(1-c\alpha^2) \lesseqgtr 0 \ .$$

Denote the left side by $h(x)$. The roots of $h(x)$ are

$$x_{1,2} = \frac{-\alpha(1-2\mu)(1-c\alpha) \pm \sqrt{\alpha^2(1-2\mu)^2(1-c\alpha)^2 - 4\alpha(1-2\mu)(1-c\alpha)\mu(1-\mu)(1-c\alpha^2)}}{2\alpha(1-2\mu)(1-c\alpha)}$$

$$= \frac{-(1-2\mu)(1-c\alpha) \pm \sqrt{(1-2\mu)^2(1-c\alpha)^2 - 4(c-1)\mu(1-\mu)(1-c\alpha^2)}}{2(c-1)\alpha} \ .$$

Note that

$$c = \frac{1}{\alpha + (1-\alpha)\log(1-\alpha)} \geq \frac{1}{\alpha - (1-\alpha)\alpha} = \frac{1}{\alpha^2} > \frac{1}{\alpha} > 1 \ ,$$

since $\log(1+x) \leq x$. This implies that the expression under the root is positive, and that

$$\sqrt{(1-2\mu)^2(1-c\alpha)^2 - 4(c-1)\mu(1-\mu)(1-c\alpha^2)} \geq |(1-2\mu)(1-c\alpha)| \ ,$$

which in turn implies that at least one of the roots of $h(x)$ is negative. Let $y = \max\{x_1, x_2\}$.

By the previous observation, the function $g_\mu(x)$ is concave on the interval $[0, y]$ and convex on the interval $[y, 1-\mu]$ (with the convention that $[a, b] = \emptyset$ if $a > b$). Noting that $g_\mu(0) = 0$ and $g'_\mu(0) = 0$ we have $g_\mu(x) \leq 0$ on $[0, y]$. On the other hand, $g_\mu(1-\mu) \leq 0 \Rightarrow g_\mu(x) \leq 0$ for $x \in [y, 1-\mu]$, by the convexity of $g_\mu(x)$ on this interval and that $g_\mu(y) \leq 0$.

Hence, all that remains to show is $g_\mu(1-\mu) \leq 0$ for all $\mu \in (0, 1)$. This yields the inequality

$$0 \geq \log\frac{1}{\mu} - c\left((\mu + \alpha(1-\mu))\log\frac{\mu+\alpha(1-\mu)}{\mu} + (1-\mu-\alpha(1-\mu))\log\frac{1-\mu-\alpha(1-\mu)}{1-\alpha}\right)$$

$$= \log\frac{1}{\mu} - c\left(((1-\alpha)\mu + \alpha)\log\left(1 + \frac{\alpha}{(1-\alpha)\mu}\right) + \log(1-\alpha)\right) := l(\mu) \ .$$

Note that the right side is equal to zero at $\mu = 1$. To conclude the inequality above, we show that the right side is increasing in $\mu$. We have

$$\frac{\partial}{\partial\mu}l(\mu) = -\frac{1}{\mu} - c\left((1-\alpha)\log\left(1 + \frac{\alpha}{(1-\alpha)\mu}\right) - \frac{\alpha}{\mu}\right)$$

$$= \frac{1}{\mu}(c\alpha - 1) - c(1-\alpha)\log\left(1 + \frac{\alpha}{(1-\alpha)\mu}\right) \ .$$

Using the inequality $\log(1+x) \leq \frac{x-a}{a+1} + \log(1+a)$ (that is the line tangential to $\log(1+x)$ at any point $a > -1$ upper bounds $\log(1+x)$) with $a = \frac{\alpha}{1-\alpha}$, we can continue as

$$\frac{\partial}{\partial\mu}l(\mu) \geq \frac{1}{\mu}(c\alpha - 1) - c(1-\alpha)\log\left(\alpha\left(\frac{1}{\mu} - 1\right) - \log(1-\alpha)\right)$$

$$= \frac{1}{\mu}(c\alpha - 1 - c\alpha(1-\alpha)) - c(1-\alpha)(\alpha - \log(1-\alpha)) \ .$$

Finally, noting that $c\alpha - 1 - c\alpha(1-\alpha)$ is positive (since $c \geq 1/\alpha^2$) we can further decrease the right hand side by using $1/\mu \geq 1$, which yields

$$\frac{\partial}{\partial\mu}l(\mu) \geq c\left(\alpha - (1-\alpha)\log(1-\alpha)\right) - 1 \ .$$

The right side is non-negative, whenever $c \geq 1/(\alpha - (1-\alpha)\log(1-\alpha))$, concluding the proof. $\quad\square$

## Proofs for Section 4

**Proposition 1.** *The lil-KLUCB algorithm is $2\delta$-PAC.*

*Proof.* Suppose that when the algorithm stops, $\text{TOP}(t) \neq 1$. This implies that there exists $t \in \mathbb{N}$ and $i \geq 2$ for which

$$L_i(f_{T_i(t)}(\delta/(n-1))) > U_1(f_{T_1(t)}(\delta)) \ .$$

Consider the events $\Omega_1(\delta)$ and $\Omega_i(\delta/(n-1))$ for $i \geq 2$, and let their intersection be

$$\Omega = \Omega_1(\delta) \cap \left(\cap_{i\geq 2}\Omega_i(\delta/(n-1))\right) \ .$$

Note that

$$\mathbb{P}(\Omega) = 1 - \mathbb{P}(\overline{\Omega}) \geq 1 - \mathbb{P}\left(\overline{\Omega}_1(\delta) \cup (\cup_{i\geq 2}\overline{\Omega}_i(\delta/(n-1)))\right) \geq 1 - 2\delta$$

by Theorem 1 (where $\overline{\Omega}$ is the complementary event of $\Omega$). However, on the event $\Omega$ the algorithm cannot fail, as on this event $L_i(f_{T_i(t)}(\delta/(n-1))) \leq \mu_i$ and $U_1(f_{T_1(t)}(\delta)) \geq \mu_1$ which (together with the first display) would imply $\mu_1 < \mu_i$, a contradiction. $\quad\square$

The backbone to proving Theorem 1 is the following lemma. Recall that for $\mu, \widetilde{\mu} \in [0,1]$, the Chernoff information $D^*(\mu, \widetilde{\mu})$ between two Bernoulli random variables with parameters $\mu$ and $\widetilde{\mu}$ can be written as

$$D^*(\mu, \widetilde{\mu}) = \inf_{x \in (0,1)} \max\{D(x, \mu), D(x, \widetilde{\mu})\} \ .$$

**Lemma 2.** *Let $Y_1, Y_2, \ldots$ be independent samples from a distribution $\mathbb{P}$, and consider a sequence of confidence bounds for the mean $\mu$ of the form*

$$U(f_t(\delta)) = \sup\{m > \widehat{\mu}_t : \ D(\widehat{\mu}_t, m) \leq f_t(\delta)\} \ ,$$

*where $\widehat{\mu}_t$ is the empirical mean based on $\{Y_j\}_{j\in[t]}$, $\delta \in (0,1)$ and $f_t(x)$ is decreasing in $x$. Consider a realization of the sequence $\{\widehat{\mu}_t\}_{t\in\mathbb{N}}$, and suppose that $\epsilon \in (0,1)$ is such that*

$$D(\widehat{\mu}_t, \mu) \leq f_t(\epsilon) \ \forall t \in \mathbb{N} \ .$$

*Then for any fixed $\widetilde{\mu} \in (\mu, 1)$ we have*

$$f_t(\delta \cdot \epsilon) < D^*(\widetilde{\mu}, \mu) \ \Rightarrow \ U(f_t(\delta)) < \widetilde{\mu} \ .$$

*Proof.* We first note that $f_t(\delta \cdot \epsilon) \geq \min\{f_t(\delta), f_t(\epsilon)\}$ since $\delta, \epsilon \leq 1$ and $f_t(\cdot)$ is decreasing.

The claim then follows by the definitions of $D^*(\mu, \widetilde{\mu}), U_t(\delta)$ and $\epsilon$. In particular, on one hand $D(\widehat{\mu}_t, \mu) \leq f_t(\delta \cdot \epsilon)$ for every $t \in \mathbb{N}$. On the other hand,

$$\widetilde{\mu} \leq U_t(\delta) \iff D(\widehat{\mu}_t, \widetilde{\mu}) \leq f_t(\delta) \Rightarrow D(\widehat{\mu}_t, \widetilde{\mu}) \leq f_t(\delta \cdot \epsilon) \ .$$

This would imply that for $\widehat{\mu}_t$ we both have both $D(\widehat{\mu}_t, \mu) \leq f_t(\delta \cdot \epsilon)$ and $D(\widehat{\mu}_t, \widetilde{\mu}) \leq f_t(\delta \cdot \epsilon)$. However, this is impossible, by the definition of $D^*(\widetilde{\mu}, \mu)$. $\quad\square$

With this lemma, we are ready to prove Theorem 1.

*Proof of Theorem 1.* Observe that at each time step two things can happen in the algorithm (apart from stopping): *(1)* Arm 1 is not pulled (two sub-optimal arms are pulled); *(2)* Arm 1 is pulled together with some other (suboptimal) arm.

Our aim is to upper bound the number of times any given arm is be played for either of the reasons above. We do so on an event of the form

$$\Omega' = \bigcap_{i \in [n]} \Omega_i(\delta_i) \,,$$

as a function of the quantities $\{\delta_i\}_{i \in [n]}$, invoking Lemma 2. We set $\delta_1 = \delta$ and choose $\{\delta_i\}_{i \geq 2}$ such that they take the largest possible values, i.e. $\delta_i = \sup\{\epsilon \in (0,1) : \Omega_i(\epsilon) \text{ holds}\}$. Finally, we control the contribution of these random $\delta_i$ to the sample complexity bound obtained in the previous step.

Note that we know from Theorem 1 that $\mathbb{P}(\overline{\Omega}_1(\delta)) \leq \delta$.

**A sample complexity bound under $\Omega'$:** If Arm 1 is not pulled at time $t$, there has to exist another Arm $i$ such that $\widehat{\mu}_{i,t} \geq \widehat{\mu}_{1,t}$. Under the event $\Omega_1(\delta)$ this can no longer happen once $U_i(f_{T_i(t)}(\delta)) < \mu_1$. By Lemma 2 the latter is guaranteed when

$$f_{T_i(t)}(\delta \cdot \delta_i) < D^*(\mu_i, \mu_1) \,.$$

Using the notation

$$\tau_i(\delta \cdot \delta_i) = \min\left\{t \in \mathbb{N} : f_t(\delta \cdot \delta_i) < D^*(\mu_i, \mu_1)\right\} \,,$$

we know that any suboptimal Arm $i$ can only be pulled at most $\tau_i(\delta \cdot \delta_i)$ times in a way that it is not pulled together with Arm 1. Hence, Arm 1 will be played eventually.

Suppose that at time $t$ a suboptimal Arm $i$ ($i \geq 2$) is pulled together with Arm 1. This can only happen if the confidence regions of the means of the two arms overlap at time $t$, i.e. $L_1(f_{T_1(t)}(\delta)) \leq U_i(f_{T_i(t)}(\delta))$. However, this is impossible once there exists a value $\widetilde{\mu}_i \in (\mu_i, \mu_1)$ that separates the two confidence bounds, i.e $U_i(f_{T_i(t)}(\delta)) < \widetilde{\mu}_i < L_1(f_{T_1(t)}(\delta))$.

According to Lemma 2, this happens once $T_i(t)$ is such that

$$f_{T_i(t)}(\delta \cdot \delta_i) \leq D^*(\mu_i, \widetilde{\mu}_i) \,,$$

and $T_1(t)$ is such that

$$f_{T_1(t)}(\delta) \leq D^*(\mu_1, \widetilde{\mu}_i) \,.$$

Note that in the second inequality, the quantity on the left hand side can indeed be chosen as $f_{T_1(t)}(\delta)$ instead of $f_{T_1(t)}(\delta^2)$), which can be easily seen by consulting the proof of Lemma 2.

For $i \geq 2$ let

$$\xi_i(\delta \cdot \delta_i) = \min\left\{t \in \mathbb{N} : f_t(\delta \cdot \delta_i) < D^*(\mu_i, \widetilde{\mu}_i)\right\} \,,$$

and

$$\xi_1(\delta) = \min\left\{t \in \mathbb{N} : f_t(\delta/(n-1)) < \min_{i \geq 2} D^*(\mu_1, \widetilde{\mu}_i)\right\} \,.$$

By monotonicity of the Chernoff-information $\xi_i(\delta \cdot \delta_i) \geq \tau_i(\delta \cdot \delta_i)$ for every $i \geq 2$. Thus, Arm $i$ can not be pulled more than $\xi_i(\delta \cdot \delta_i)$ times.

Hence the sample complexity on the event $\Omega'$ is upper bounded by

$$\xi_1(\delta) + \sum_{i \geq 2} \xi_i(\delta \cdot \delta_i) \,.$$

**Controlling the contribution of the $\delta_i$:** It is easy to check that there exists a universal constant $K_1$ such that

$$\xi_i(\delta \cdot \delta_i) \leq \frac{K_1 \log\left((\delta \cdot \delta_i)^{-1} \log D^*(\mu_i, \widetilde{\mu}_i)^{-1}\right)}{D^*(\mu_i, \widetilde{\mu}_i)} \,.$$

and

$$\xi_1(\delta) \leq \frac{K_1 \log\left((n-1)\delta^{-1} \log D^*(\mu_1, \widetilde{\mu})^{-1}\right)}{D^*(\mu_1, \widetilde{\mu})} \,.$$

Now let $\delta_i = \sup\{\epsilon > 0 : U_i(f_t(\epsilon)) \geq \mu_i \; \forall t \in \mathbb{N}\}$. We have

$$\mathbb{P}(\delta_i < \gamma) = \mathbb{P}(\exists t \in \mathbb{N} : U_i(f_t(\epsilon)) \geq \mu_i) \leq \gamma$$

according to Theorem 1. Hence, substituting $\gamma = \exp(-D^*(\mu_i, \widetilde{\mu}_i)z)$ we get

$$\mathbb{P}\left(\frac{\log \delta_i^{-1}}{D^*(\mu_i, \widetilde{\mu}_i)} \geq z\right) \leq \exp(-D^*(\mu_i, \widetilde{\mu}_i)z) \,.$$

Hence $D^*(\mu_i, \widetilde{\mu}_i)^{-1} \log \delta_i^{-1}$ are independent sub-exponential random variables. Using standard techniques for bounding sums of sub-exponential random variables, we have

$$\mathbb{P}\left(\sum_{i \geq 2} \frac{\log \delta_i^{-1}}{D^*(\mu_i, \widetilde{\mu}_i)} \geq K_2 \sum_{i \geq 2} \frac{\log \delta^{-1}}{D^*(\mu_i, \widetilde{\mu}_i)}\right) \leq \delta \,,$$

with some constant $K_2$.

Combining this inequality with those for $\xi_i(\cdot)$ concludes the proof. $\qquad\square$

## The sub-Gaussian tail-bounds for the numerical comparisons of Section 5

We can get a sub-Gaussian tail bound as well with the method of Theorem 2 as follows. We start by the same union-bound 4.

Upper bounding the terms in the second sum go analogously up to the display 6. At that point, we can use Pinsker's inequality stating that $2(x - y)^2 \leq D(x, y)$ (see [1])[1]. This yields

$$\mathbb{P}\left(\exists t \in [2^k, 2^{k+1}] : \; \widehat{\mu}_t - \mu > z_t\right) \leq \exp\left(-2t_j \left(\frac{N+j-1}{N+j}\right)^2 z_{t_{j-1}}^2\right) \,.$$

Recall that $t_j = (1 + \frac{j}{N})2^k$ and define

$$z_t = \sqrt{\frac{1}{2}\left(\frac{N+1}{N}\right)^2 \frac{\log\left(\kappa(N)\log_2(2t)/\delta\right)}{t}} \,,$$

where $\kappa(N)$ is the same constant as in the statement of Theorem 2. Note that the sequence $tz_t$ is increasing, which was required for the computations leading to 6.

Plugging in these values, we get

$$\mathbb{P}\left(\exists t \in [2^k, 2^{k+1}] : \; \widehat{\mu}_t - \mu > z_t\right)$$
$$\leq \exp\left(-\frac{N+j-1}{N+1}\left(\frac{N+1}{N}\right)^2 \log\left(\kappa(N)\log_2\left(2^{k+1}\frac{N+j}{N}\right)/\delta\right)\right)$$
$$\leq \delta^{\frac{N+1}{N}}\kappa(N)^{-\frac{N+1}{N}}(k+1)^{-\frac{N+1}{N}} \,,$$

where the last line follows by $j \geq 1$.

As for the first term in 4 we can also use Pinsker's inequality to get

$$\mathbb{P}(\exists t \in [N] : \; \widehat{\mu}_t - \mu > z_t) \leq \exp\left(-\left(\frac{N+1}{N}\right)^2 \log\left(\kappa(N)\log_2(2t)/\delta\right)\right)$$
$$\leq \delta^{\frac{N+1}{N}}\kappa(N)^{-\frac{N+1}{N}} \sum_{t \in [N]} \log_2(2t)^{-\frac{N+1}{N}} \,.$$

The proof concludes the same way as that of Theorem 2, so that with the definition of $z_t$ above we have that

$$\mathbb{P}(\exists t \in \mathbb{N} : \; \widehat{\mu}_t - \mu > z_t) \leq \delta \,.$$

## The New Yorker Cartoon Caption Contest

Each week a cartoon in need of a caption appears in The New Yorker magazine. The readers are invited to submit their ideas for funny captions to go with that cartoon. The New Yorker selects three finalists from the submissions, after which the readers select their favorite by voting online at `http://contest.newyorker.com/CaptionContest.aspx?tab=vote`.

Please rank the entries for this Cartoon Caption Contest image, then click the "Done" button. You can rank as many or as few captions as you like, but five is too few and five thousand is way too many.

"*I asked Ted to drop in on this meeting.*"

| UNFUNNY | SOMEWHAT FUNNY | FUNNY |
|---------|----------------|-------|

DONE

## Footnotes

[1]Note that another approach would be to use Hoeffding's bound for the moment generating function $E(e^{\lambda(Y_1 - \mu)})$ at 5. At the end, this would result in the same result as using Pinsker's inequality.

# References

[1] Alexandre B. Tsybakov. *Introduction to Nonparametric Estimation*, volume 41 of *Mathématiques & Applications (Berlin) [Mathematics & Applications]*. Springer, Berlin, 2009.