[Reviews · NeurIPS 2017]

Reviewer 1



The paper proposes a novel PAC algorithm for the multi-armed bandit problem that combines ideas from KL-LUCB and lil-UCB. For the proposed algorithm (called lil-KLUCB) the authors show a sample complexity upper bound that improves on the corresponding result of both its predecessors. The core of this result is a novel concentration bound which uses the Chernoff-information - similiarly as in case of KL-LUCB, but this new approach results in a much tighter bound. Finally, the paper presents some experimental results demonstrating that the theoretical improvement has practical implications as well. The problem I have found with the paper is that the proof of Theorem 1 contains errors that make it completely confusing, and thereby the correctness of the results remains unclear. What is exactly A_k and t_j? Their current definition do not makes sense: why should t in the definition of A_k run over the whole set of natural numbers? And why should t_k be bigger than, but still proportional to 2^k? And which part of the proof corresponds to the first term on the RHS of the equation below line 135, and which part to the rest of the terms? Additionally, the proof sketch of Theorem 1 is is rather messy too (unfortunately, I was not able to check the corresponding part in the supplementary material): - In line 180, what is U(f_t(\epsilon))? It cannot be deduced from the definition of U(t,\epsilon). - Still in line 180, the equation in the second half of the line does not seem right either: the first term depends on gamma, whereas the second term depends on \epsilon. What exactly is going on there? - In line 181 you are referring to some result from Theorem 1 - however, you are proving Theorem 1. (Presumably this reference is to the theorem in the supplementary.) Nevertheless, the results are nice and plausible, and imprive significantly on previous bounds (although they are still somewhat worse than existing lower bounds); in fact, the novel concentration bound could be of interest on its own. Therefore, I would recommend acceptance given the issues discussed above could be taken care of in a convincing fashion. Further remarks and questions: - There does not seem to be a definition for D(z,x). - How do you calculate L_i(t,\epsilon) and U_i(t,\epsilon)? - In line 126: it seems that 1-\delta should be 1-2\delta. - Line 142: the left bracket is missing in the term "D\mu,\mu)". - In the line before line 142: "exp(\lambda S_t)" should be "exp(\lambda S_{t_j})". - In the line below 135: S_t is not yet defined. - In lines 194 and 195 you are referring to Fig. 5, whereas you only have Fig. 1. - How close do you think the bounds are to the actual optimum of the problem?

Reviewer 2



The authors introduce a new algorithm for simple regret minimization with a sample complexity which combines the better dependency of KL-UCB on the arms’ value and the better dependency of lil-UCB on the number of arms. The authors provide an analysis of the sample complexity and experiments with real-world data. The contribution is clearly stated: The authors properly explain in Section 1 the advantages of KL-UCB and lil-UCB. In Section 2, they introduce the KL-LUCB algorithm which is simple and practical (total computational time is linear with the number of arms pulled). The insights on the advantage of KL-UCB (with a low number of arms) and KL-LUCB over lil-UCB are clear even though I have a question: How is the last bound on the Chernoff information before line 86 obtained? This could be more detailed. (maybe in the appendix?). Is it possible to easily compute kappa(N) ? It seems that by using kappa(N) and the value of the second sum in the definition of kappa(N) (Thm2) it is possible to compute kappa(N+1) in constant computational time. Is this the case? In any case, I think the authors should provide a way to get the value of kappa and c in the algorithm description so that the people could directly implement the algorithm without digging in the proof. Under line 54 definition of U_i : inf -> sup Line 56: I think your bound is even stronger and applies with probability 1-epsilon simultaneously for all t. If this is the case, you should mention it (exchange “for all t” with “with prob 1-eps” and replace mu_i by mu_i,t). Line 88 of the appendix: “It is easy to check that… “ is indeed easy but kind of long computation involving computing a bound on kappa, … Adding some steps there would make the proof easier to check. AFTER REBUTTAL I have read the rebuttal and I am satisfied with the answers of the authors. After considering other reviews, I reduced the review score due to lack of clarity in the technical parts of the paper. Yet, I still think that the main results and the paper are strong and deserve attention.

Reviewer 3



This paper propose a new algorithm for the problem of best-arm identification with fixed confidence in multi-armed bandits. The new proposed algorithm "lil-KLUCB" is a marriage of two recent algorithms: "lil-UCB" and "KL-LUCB". According to the authors, the key improvement from these algorithms is: - not being based on sub-Gaussian confidence bounds; - to use Chernoff information for problem-dependent bounds instead of KL-divergence (or square gap); - to tightly exploit the assumption of bounded rewards; - to remove a logarithmic term in the number of arms. The problem-dependent sample complexity bound for d-correct best arm identification is in K.log(log(1/D)/d)/D for "lil-KLUCB" instead of K.log(K/(dD))/D for "KL-LUCB". This is an improvement when the number of arms K is large. To illustrate the algorithm's analysis, a few simulations are performed based on crowd-sourcing data against KL-LUCB and lil-UCB. The scientific contribution (improving KL-type bounds on best-arm identification problem) is incremental but the theoretical work seem solid. The "strong motivation" toward "large scale Crowdsourcing" sounds a bit overplayed. Remark: The literature on best-arm identification is already quite rich. To name a few existing algorithms, we have: lil-ucb, kl-lucb, kl-racing, exponential-gap elimination, sequential halving, median elimination, successive elimination, lucb1, and prism. On that ground it would make sense to add a few more competitors in the experiments. AFTER REBUTTAL Reading the rebuttal I was quite surprised to learn that the New-Yorker (through the "NEXT" API) chose such a fresh MAB algorithm to run their caption contest. It could be worth adding a reference to their white paper: http://nextml.org/assets/next.pdf